Classification of the stages of Alzheimer’s disease based on three-dimensional lightweight neural networks

http://orcid.org/0000-0002-2478-2997 Li Jun
Liu Juntong liu.juntong@wnmc.edu.cn
Su Yang
Chang Jie
Ye Mingquan
School of Medical Information, Wannan Medical College , Wuhu , China
Wang Wei
Electronic publication date: 2025 May 15
Publication date: 2025
Volume: 11
Electronic Location ID: e2897
Received 2024 Nov 14; Accepted 2025 Apr 24
Copyright: © 2025 Li et al.
Copyright year: 2025
Copyright holder: Li et al.
License: This is an open access article distributed under the terms of the Creative Commons Attribution License, which permits unrestricted use, distribution, reproduction and adaptation in any medium and for any purpose provided that it is properly attributed. For attribution, the original author(s), title, publication source (PeerJ Computer Science) and either DOI or URL of the article must be cited.
License URL: https://creativecommons.org/licenses/by/4.0/

Keywords: Alzheimer’s disease, Classification, Magnetic resonance imaging, Three-dimensional lightweight neural networks

Funding: Key Research for Humanities and Social Sciences of Anhui Province, China 2023AH051726 Young and Middle-Aged Scientific Research Fund of Wannan Medical College WK202209 and WK2024ZQNZ19 Doctoral Research Foundation of Wannan Medical College WYRCQD2021002 University Synergy Innovation Program of Anhui Province, China GXXT-2022-044 Key Research and Development Plan of Anhui Province, China 2022a05020011 Excellent Scientific Research Innovation Team Project of Universities in Anhui Province, China 2022AH010075 Research project of the Institute of Bee Research, Chinese Academy of Agricultural Sciences H202406 This work was supported by the Key Research for Humanities and Social Sciences of Anhui Province, China (Grant No. 2023AH051726), the Young and Middle-Aged Scientific Research Fund of Wannan Medical College (Grant No. WK202209 and WK2024ZQNZ19), the Doctoral Research Foundation of Wannan Medical College (Grant No. WYRCQD2021002), the University Synergy Innovation Program of Anhui Province, China (Grant No. GXXT-2022-044), the Key Research and Development Plan of Anhui Province, China (Grant No. 2022a05020011), the Excellent Scientific Research Innovation Team Project of Universities in Anhui Province, China (Grant No. 2022AH010075), and the Research project of the Institute of Bee Research, Chinese Academy of Agricultural Sciences (Grant No. H202406). There was no additional external funding received for this study. The funders had no role in study design, data collection and analysis, decision to publish, or preparation of the manuscript.

==============================
Alzheimer’s disease is a neurodegenerative disease that seriously threatens the life and health of the elderly. This study used three-dimensional lightweight neural networks to classify the stages of Alzheimer’s disease and explore the relationship between the stages and the variations of brain tissue. The study used CAT12 to preprocess magnetic resonance images of the brain and got three kinds of preprocessed images: standardized images, segmented standardized gray matter images, and segmented standardized white matter images. The three kinds of images were used to train four kinds of three-dimensional lightweight neural networks respectively, and the evaluation metrics of the neural networks are calculated. The accuracies of the neural networks for classifying the stages of Alzheimer’s disease (cognitively normal, mild cognitive impairment, Alzheimer’s disease) in the study are above 96%, and the precisions and recalls of classifying the three stages are above 94%. The study found that for the classification of cognitively normal, the best classification results can be obtained by training with the segmented standardized gray matter images, and for mild cognitive impairment and Alzheimer’s disease, the best classification results can be obtained by training with the standardized images. The study analyzed that in the process of cognitively normal to mild cognitive impairment, variations in the segmented standardized gray matter images are more obvious at the beginning, while variations in the segmented standardized white matter images are not obvious. As the disease progresses, variations in the segmented standardized white matter images tend to become more significant, and variations in the segmented standardized gray matter images and white matter images are both significant in the development of Alzheimer’s disease.

Introduction

Alzheimer’s disease (AD) is a life-threatening neurodegenerative disease common in the elderly. The disease manifests itself in a series of symptoms, such as memory loss and cognitive impairment, which seriously affect the patient’s daily life until death. To our knowledge, there are no drugs or other medical treatments that can cure AD, but studies have found that if AD can be diagnosed in the early stage, some drugs and treatments could slow the progression of AD (Rasmussen & Langerman, 2019). Therefore, the early diagnosis of AD is particularly important and meaningful. The early stages before AD are cognitively normal (CN) and mild cognitive impairment (MCI) (Storandt, 2008). At CN, individuals show no noticeable symptoms of cognitive decline. Brain variations associated with Alzheimer’s disease may begin, but they do not impact daily functioning. Individuals with MCI experience noticeable memory or cognitive issues that are greater than expected for their age, but not severe enough to interfere significantly with daily life. MCI is often considered a risk factor for developing Alzheimer’s disease, but not everyone with MCI will go on to develop the disease.

Currently, there are many methods for the early diagnosis of AD. Clinical evaluations include cognitive function tests, such as the Mini-Mental State Examination (MMSE) and the Montreal Cognitive Assessment (MoCA) (Pinto et al., 2019). Imaging examinations include magnetic resonance imaging (MRI) to detect variations in brain structures, such as hippocampal atrophy and ventricular enlargement (Frisoni et al., 2010). Positron emission tomography (PET) could show the accumulation of amyloid plaques in the brain (Nordberg et al., 2010). Biomarker tests include analysis of cerebrospinal fluid to detect changes in amyloid and tau protein levels (Olsson et al., 2016). As a non-invasive examination method, MRI is widely used in the early diagnosis of AD. There are a lot of related works that used MRI to diagnose the early stage of Alzheimer’s disease with machine learning methods. Raghavaiah & Varadarajan (2021) used the squirrel search algorithm to optimize the deep neural network to classify the preprocessed gray matter part of MRI. Tuan et al. (2022) first used a Gaussian mixture model with a convolutional neural network (CNN) to segment the MRI into gray matter (GM), white matter (WM), and cerebrospinal fluid (CSF). Then, CNN was used to extract the features of the segmented images, and extreme gradient boosting (XGBoost) was used for feature selection. Finally, a support vector machine (SVM) was used for classification. Sarraf & Tofighi (2017) used LeNet and GoogleNet to classify the AD stages with GM MRI. Basheera & Ram (2020, 2019, 2021) used enhanced independent component analysis to get the segmented GM from T2-weighted MRI and then used the segmented GM MRI to train a CNN for classifying AD stages. Ji et al. (2019) used the segmented GM and WM images to train a CNN and improved the accuracy of classification through ensemble learning. Ortiz et al. (2017) used sparse inverse covariance estimation to obtain features from the segmented GM and WM MRI, and then used these features as input to train a deep-learning model to classify AD stages. Houria et al. (2022) fused the anisotropy fraction and mean diffusion coefficients in diffusion tensor imaging (DTI) with the GM MRI as multimodal features, and that were used by SVM for classification. Maqsood et al. (2019) used the segmented GM, WM, CSF MRI, and unsegmented complete MRI to fine-tune the pre-trained AlexNet to make it suitable for classifying AD stages. Huang et al. (2022) used voxel-based morphology to get the segmented GM MRI and used GM MRI to train a CNN for classification. All of the above studies achieved good accuracies of classification at those times. There are also many other studies using MRI to classify AD stages for the early diagnosis of AD (Khvostikov et al., 2018; Yagis et al., 2020; Folego et al., 2020; Hazarika et al., 2023; Li & Yang, 2021; Turkan & Tek, 2021). These studies treated MRI as a whole and did not segment MRI into the GM and WM images. The two models recently designed by Ozdemir & Dogan (2024a, 2024b) can diagnose brain tumors and Alzheimer’s disease, respectively. The methylthioadenosine phosphorylase (MTAP) model combines Adaptive Synthetic Sampling Approach for Imbalanced Learning (ADASYN) oversampling, network pruning, and Avg-TopK pooling with DenseNet201 to achieve 99.69% accuracy in brain tumor classification (glioma, meningioma, pituitary tumor), optimizing computational efficiency and diagnostic focus on temporal/parietal regions (Ozdemir & Dogan, 2024a). A custom CNN integrating Squeeze-and-Excitation (SE) blocks, SMOTE oversampling, and Avg-TopK pooling enables 99.84% accuracy in Alzheimer’s disease diagnosis, with Grad-CAM visualization highlighting cortical pathology-related features, surpassing traditional models like VGG and ResNet (Ozdemir & Dogan, 2024b).

The novelty and contribution of this study are mainly reflected in the following two aspects: Application of 3D CNNs for early diagnosis: While many previous studies have utilized 2D imaging techniques for AD diagnosis, our research employs 3D CNNs to extract features from 3D brain images. This approach allows for a more comprehensive capture of spatial information, leading to enhanced diagnostic accuracy. Recent studies have demonstrated that 3D CNNs can effectively identify AD biomarkers from structural MRI (sMRI) scans (Folego et al., 2020).

Integrated analysis of disease staging and brain tissue impact: Beyond focusing solely on diagnostic accuracy, our study concurrently classifies disease stages and investigates the effects of AD on brain tissues, specifically gray matter, and white matter. This dual approach provides deeper insights into the pathological progression of AD.

The structure of the article is as follows: The next section outlines the materials and methods used in this study, followed by a presentation of the results and discussion. Finally, the conclusions are summarized.

Materials and Methods

Dataset

The dataset for this study was downloaded from the public dataset Alzheimer’s Disease Neuroimaging Initiative (ADNI) T1-weighted Magnetization Prepared Rapid Acquisition Gradient Echo (MPRAGE) MRI (Jack et al., 2008). MPRAGE is an MRI sequence used for brain imaging, especially in neuroimaging, which is widely used to display brain structures in detail, such as the boundary between GM and WM. MPRAGE is often used to detect pathological variations in the brain, such as brain atrophy in AD, which is why this MRI sequence was used in this study. It should be noted that since the structure of the brain varies over time, in this study, images collected from the same person at different times are considered different images. As shown in Table 1, the statistical information of the dataset has been given.

Table 1 Statistical information of the dataset.

Age	Gender	AD stages	
	Male	Female	CN	MCI	AD	
76.34±7.07	1,896	1,627	1,530	1,275	718	

The downloaded original images are single-layer Digital Imaging and Communications in Medicine (DICOM) format images, which need to be converted into 3D images in the Neuroimaging Informatics Technology Initiative (NIfTI) format for subsequent processing. DICOM is a format widely used in medical imaging that stores both image data and metadata, including patient details and imaging parameters. NIfTI is a format primarily used for storing and sharing brain imaging data, particularly in neuroimaging research. It focuses on storing volumetric or multidimensional data with a simpler structure compared to DICOM, often used for MRI data. There are mainly the following four steps using CAT12 (https://neuro-jena.github.io/cat/) to preprocess MRI: spatial normalization, bias field correction, tissue classification, and modulation. Spatial normalization: The MRI image is then spatially normalized to a standard template (such as the MNI template (Lancaster et al., 2007)). This step aligns the individual scan to a common anatomical space, allowing for consistent tissue segmentation across subjects. The normalization process involves both affine and non-linear transformations, ensuring that anatomical differences between subjects are accounted for.

Bias field correction: MRI scans often exhibit low-frequency intensity variations due to magnetic field inhomogeneities. To correct this, CAT12 applies a bias field correction, which reduces the impact of these inhomogeneities and enhances the accuracy of tissue classification.

Tissue classification: The process begins by classifying the MRI scan into GM, WM, and CSF based on a probabilistic model. The intensity of each voxel in the image is compared to a predefined tissue probability map, which reflects the typical distribution of tissue intensities across the brain. The classification is not solely intensity-based; it incorporates spatial information to improve accuracy, especially in areas where tissue boundaries are ambiguous.

Modulation: During spatial normalization, each individual’s brain is transformed to match a standard template. This transformation involves scaling, rotation, and warping, which may stretch or compress the brain. Modulation compensates for these local changes by scaling the tissue maps to preserve the original tissue volume. Without modulation, the segmented tissue maps would only reflect the spatial distribution of tissue types, not their actual volume.

Spatial normalization, bias field correction, and modulation are called standardization in this study. The final output consists of separate tissue probability maps for GM, WM, and CSF. CAT12 includes an automated quality control mechanism for segmentation. This feature provides quantitative metrics that assess the accuracy of the segmentation process, ensuring that the outputs meet the required standards without the need for extensive manual intervention. Furthermore, we chose CAT12 over other segmentation tools because of its robust performance, seamless integration with SPM (https://www.fil.ion.ucl.ac.uk/spm/), and its extensive validation in numerous neuroimaging studies, which together ensure high reliability and reproducibility of the segmentation results. This study mainly studies AD from a structural perspective, so the standardized images, as well as the segmented standardized GM and WM images, are mainly used. The resolution of the images is 113×113×137 (corresponding to depth × height × width). As shown in Fig. 1, the standardized image, as well as the segmented GM and WM images are given. The standardized image is represented by wm (i.e., warped and modulated), the segmented GM image is represented by mwp1, and the segmented WM image is represented by mwp2.

Figure 1 (A) The standardized image (wm), (B) the segmented GM image (mwp1), (C) the segmented WM image (mwp2).

Methods

Since brain tissues are 3D structures, to learn the 3D spatial information of MRI, the study decided to use 3D neural networks to learn the features of images. The neural networks used in this study are 3D-SqueezeNet, 3D-MobileNetV1, and 3D-ShuffleNetV1. As a reference, the 3D version of the classic neural network ResNet18 is also used. The code of Köpüklü et al. (2019) was modified for this study. Due to the small scale of the dataset, to avoid overfitting of training, this study uses lightweight neural networks. The common point of these neural networks is that the depth of the neural networks is relatively shallow, the parameters of the neural networks are relatively small, and while the parameters of neural networks remain small, the performance of the neural networks is not much lower than that of deeper neural networks. Since these lightweight neural networks have been described in detail in relevant literature (Köpüklü et al., 2019), here is just a brief description of these lightweight neural networks.

The 3D-SqueezeNet architecture begins with a convolutional layer (Conv1) that performs downsampling with a stride of (1,2,2). This is followed by eight Fire modules (Fire2-9), each of which consists of a 1×1 Squeeze layer and a 3×3 Expand layer. Finally, there is a convolutional layer (Conv10) with a stride of (1,1,1), followed by an average pooling layer and a linear layer for classification. The original model takes as input a 16-frame clip with a spatial resolution of 112×112 pixels. The accuracy of SqueezeNet has reached the level of the classic AlexNet, while the parameters of the neural network are reduced by 50 times compared to AlexNet. The parameter of 3D-SqueezeNet is 1839299 in the experiment.

The 3D-MobileNetV1 architecture starts with a convolutional layer (Conv1) that downsamples with a stride of (1,2,2). Subsequently, it includes 13 MobileNet blocks, where each block comprises a depthwise convolution (DWConv) and a pointwise convolution ( 1×1 convolution). The output of the model is processed through an average pooling layer, followed by a linear layer for classification. The input to the original model is a 16-frame clip with a spatial resolution of 112×112 pixels. The parameter of 3D-MobileNetV1 is 3301059 in the experiment.

The 3D-ShuffleNetV1 architecture commences with a convolutional layer (Conv1) that downsamples using a stride of (1,2,2). It is then followed by 16 ShuffleNet blocks, which are grouped into three stages. Each ShuffleNet block includes a pointwise group convolution (GConv) and a channel shuffle operation. The model’s output passes through an average pooling layer, followed by a linear layer for classification. The original input is a 16-frame clip with a spatial resolution of 112×112 pixels. The parameter of 3D-ShuffleNetV1 is 947787 in the experiment.

In addition to the above three lightweight neural networks, this study also used the 3D version of the classic ResNet18. The 3D-ResNet18 architecture initiates with a convolutional layer (Conv1) that downsamples with a stride of (1,2,2). This is succeeded by 17 ResNet blocks, each of which employs a bottleneck design and includes 1×1, 3×3, and 1×1 convolutional layers, utilizing identity shortcut connections for dimension matching. The output of the model goes through an average pooling layer, followed by a linear layer for classification. The original model input is a 16-frame clip with a spatial resolution of 112×112 pixels. The parameter of 3D-ResNet18 is 33031619 in the experiment.

The experimental dataset is randomly divided into two parts: training set and testing set, of which the training set accounts for 80% and the testing set accounts for 20%. To alleviate the problem of class imbalance to some extent, the proportion of AD stages in the training set and the testing set is consistent with the proportion in the original dataset. The numbers of CN, MCI, and AD in the training set are 1,224, 1,020, and 574, respectively. The numbers of CN, MCI, and AD in the testing set are 306, 255, and 144, respectively. Subsequently, we plan to adopt additional data augmentation techniques to enhance the model’s performance; for example, the RGB-Angle-Wheel method proposed by Ozdemir, Dogan & Kaya (2024). The experiment uses wm, mwp1, and mwp2 images to train the above four 3D neural networks, and calculates the evaluation metrics of the neural networks on the testing set. Finally, the evaluation metrics of these neural networks trained with different kinds of images (wm, mwp1, and mwp2) are compared and discussed. It should be noted that to evaluate the neural networks fairly, the neural networks have never seen the testing set before calculating the evaluation metrics on the testing set.

The confusion matrix is an important tool for evaluating neural networks of multi-classification. Each row of the confusion matrix represents the actual class, while each column represents the predicted class. The diagonal of the confusion matrix represents all correctly predicted instances. This study is a problem of multi-classification, and it is needed to give the evaluation metrics of each class. For example, for CN, the instance is called a true positive (TP) when that is CN actually and the neural network predicts that is CN, the instance is called a false negative (FN) when that is CN actually and the neural network predicts that is not CN, the instance is called a false positive (FP) when that is not CN actually and the neural network predicts that is CN, and the instance is called a true negative (TN) when that is not CN actually and the neural network predicts that is not CN. Precision is defined as

(1) Precision=TPTP+FP.

Precision refers to the predicted results, which is the ratio of samples that are true positive TP to samples predicted as positive TP + FP. The higher the precision, the more reliable the prediction as positive is. Recall is defined as

(2) Recall=TPTP+FN.

The recall refers to the samples themselves and is the ratio of correctly predicted true positive samples TP to the total positive samples TP + FN. The higher the recall, the lower the probability of a missed diagnosis. It should be pointed out that, for CN, the precision should be as high as possible. The higher the precision, the more accurate and reliable the prediction as CN is, and the lower the rate of misdiagnosis is. For the two disease stages MCI and AD, the recall should also be as high as possible, and the missed diagnosis rate should be as low as possible for the stages of the disease.

Results and discussion

The training and testing of these neural networks were performed on a Linux server. The server hardware configurations are Intel® Xeon® Gold 5218R CPU @ 2.10 GHz (4 CPUs), 512 GB memory, and four NVIDIA GeForce RTX 3090 graphics cards with 24 GB video memory. The experiment employs a carefully designed training procedure using stochastic gradient descent (Zaras, Passalis & Tefas, 2022) with categorical cross-entropy loss. An initial learning rate of 0.04 is set and reduced several times by a factor of 0.1 when the loss converges. A mini-batch size of approximately eight is used, along with momentum and dampening both set to 0.9 and a weight decay of 1×10−3. To mitigate overfitting, dropout is applied before the final convolution or linear layer, with a rate of 0.2. The detailed model hyperparameters are listed in Table 2. As shown in Fig. 2, the loss functions vary with the epochs given. It can be seen from the figure that after 250 epochs, the neural networks have been convergent. From Fig. 2, it could also be found that the neural networks are trained fastest with the mwp1 images, followed by the mwp2 images, and the neural networks are trained slowest with the wm images. Explanations about this phenomenon are given here. The neural networks are trained fastest with the mwp1 means that the features of mwp1 images are most easily learned by the neural networks to classify the stages of AD, followed by the mwp2 images. Standardized images contain both gray matter and white matter parts, and the wm images are equivalent to multimodal images. Generally, it is more difficult to learn deep features from multimodal images (i.e., the wm images), but once trained, the robustness of the trained neural networks with the wm images should be better. As shown in Fig. 3, the curves of precisions with epochs are also given. It could also be observed that the precisions of the neural networks trained with the mwp1 images increased fastest, followed by mwp2 images, and the precisions of the neural networks trained with the wm images increased slowest. This similar phenomenon also means that the variations of GM features are more obvious than those of WM features in the development of AD. The neural networks can classify CN, MCI, and AD with the mwp1 images well.

Table 2 Model hyperparameters.

Hyperparameter	Value	Description	
Optimizer	SGD	Stochastic gradient descent	
Loss function	Categorical cross-entropy	Standard loss for classification tasks	
Learning rate	0.04	Initial learning rate	
Momentum	0.9	Momentum term for SGD	
Dampening	0.9	Dampening factor for momentum	
Weight decay	1×10−3	Regularization parameter to prevent overfitting	
Batch size	8	Maximum fitting mini-batch size	
Dropout rate	0.2	Dropout before the final layer to reduce overfitting	

Figure 2 (A–D) Correspond to the curves of loss functions with epochs of SqueezeNet, MobileNetV1, ShuffleNetV1, and ResNet18, respectively.

Figure 3 (A–D) Correspond to the precision curves with epochs of SqueezeNet, MobileNetV1, ShuffleNetV1, and ResNet18, respectively.

As shown in Fig. 4, the confusion matrixes of the predicted results on the testing set of each neural network trained with wm, mwp1, and mwp2 images are given. After calculation, the accuracies (i.e., the sum of the diagonal elements of the confusion matrix divided by the total number of testing sets 705) of each neural network on the testing set are above 96%. The results show that the 3D neural networks could learn deep features from the 3D images and classify the stages of AD well.

Figure 4 The confusion matrixes of the trained neural networks on the testing set.

The first column is the neural networks trained with the wm images, the second column is the neural networks trained with the mwp1 images, and the third column is the neural networks trained with the mwp2 images. Each row is a neural network.

According to the confusion matrixes that are given in Fig. 4, the precisions and recalls of each stage of AD are calculated respectively. Table 3 and Fig. 5 respectively give the precisions of classifying AD stages by the neural networks trained with different kinds of images and their corresponding histograms. From Table 3, it could be seen that the precisions of classifying AD stages of each neural network are above 94%. As shown in Fig. 5, for CN, except for ShuffleNetV1, that trained with the wm images has the highest precision, while the other three neural networks trained with the mwp1 images have the highest precisions. Overall, the precisions of the neural networks trained with the mwp1 images are the highest at CN. For MCI, the precisions of MobileNetV1 and ShuffleNetV1 trained with the wm images are the highest, the precision of SqueezeNet trained with the mwp2 images is the highest, and the precision of ResNet trained with the mwp1 or mwp2 images is the highest. Overall, for MCI, the precisions of neural networks trained with the mwp1 images are no longer the highest, but the neural networks trained with the wm or mwp2 images have the highest precisions. This phenomenon shows that the variations of WM features at MCI gradually play a significant role in the classification. For AD, except for SqueezeNet, that trained with the mwp2 images has the highest precision, while the other three neural networks trained with the wm images have the highest precisions. Overall, the precisions of the neural networks trained with the wm images are the highest at AD.

Table 3 The precisions of the neural networks trained with different kinds of images for classifying AD stages.

Networks	Images	CN	MCI	AD	
Squeezenet	wm	0.97068404	0.95703125	0.95774648	
	mwp1	0.97741935	0.96442688	0.95774648	
	mwp2	0.97106109	0.98	0.97222222	
Mobilenetv1	wm	0.96805112	0.98393574	0.98601399	
	mwp1	0.97096774	0.96031746	0.95104895	
	mwp2	0.96214511	0.98373984	0.98591549	
Shufflenetv1	wm	0.9775641	0.98790323	0.96551724	
	mwp1	0.96496815	0.97560976	0.95172414	
	mwp2	0.96784566	0.97188755	0.94482759	
Resnet	wm	0.96141479	0.97233202	0.9858156	
	mwp1	0.96507937	0.97590361	0.9787234	
	mwp2	0.96496815	0.97590361	0.97887324	

Figure 5 The histograms of precisions of the neural networks trained with different kinds of images for classifying AD stages.

(A) SqueezeNet precision; (B) MobileNet V1 precision; (C) ShuffleNetV1 precision; (D) ResNet precision.

At the stage of CN, the GM features that have been learned by the neural networks could classify CN well. The variations of WM features could be less obvious than the variations of GM features at CN, and the precisions of the neural networks trained with the mwp2 images are worse than those trained with the mwp1 images. According to the precision of MCI, the results of the neural networks trained with the wm images are better than those trained with the mwp1 images at MCI. In the process of CN to MCI, the variations of WM features gradually become significant, while the variations of GM features remain important. At the stage of AD, it could be considered that both GM and WM features have undergone obvious variations, and the precisions of the neural networks trained with the wm images are the best.

Table 4 and Fig. 6, respectively give the recalls and corresponding histograms of the neural networks trained with different kinds of images for classifying AD stages. From Table 4, we can see that the recalls of each neural network for classifying AD stages are all above 94%. As shown in Fig. 6, for CN, SqueezeNet and ResNet trained with the mwp1 images have the best recalls, MobileNetV1 trained with the mwp2 images has the best recall, and ShuffleNetV1 trained with the wm images has the best recall. Overall, the neural networks trained with the mwp1 images have the best recalls at CN. For MCI, except for SqueezeNet, where the recalls of SqueezeNet trained with the wm or mwp2 images were the best equally, the recalls of the other three neural networks trained with the wm images were the best. Overall, for MCI, the recalls of the neural networks trained with the wm images were better. For AD, except for SqueezeNet, where the recall of SqueezeNet trained with the mwp2 images was the best, the recalls of the other three neural networks trained with the wm images were the best. Overall, for AD, the recalls of the neural networks trained with the wm images were better.

Table 4 The recalls of the neural networks trained with different kinds of images for classifying AD stages.

Networks	Images	CN	MCI	AD	
squeezenet	wm	0.97385621	0.96078431	0.94444444	
	mwp1	0.99019608	0.95686275	0.94444444	
	mwp2	0.9869281	0.96078431	0.97222222	
mobilenetv1	wm	0.99019608	0.96078431	0.97916667	
	mwp1	0.98366013	0.94901961	0.94444444	
	mwp2	0.99673203	0.94901961	0.97222222	
shufflenetv1	wm	0.99673203	0.96078431	0.97222222	
	mwp1	0.99019608	0.94117647	0.95833333	
	mwp2	0.98366013	0.94901961	0.95138889	
resnet	wm	0.97712418	0.96470588	0.96527778	
	mwp1	0.99346405	0.95294118	0.95833333	
	mwp2	0.99019608	0.95294118	0.96527778	

Figure 6 The histograms of recalls of the neural networks trained with different kinds of images for classifying AD stages.

(A) SqueezeNet precision; (B) MobileNet V1 precision; (C) ShuffleNetV1 precision; (D) ResNet precision.

The discussions about the recalls of the neural networks trained with different kinds of images for classifying AD stages are almost the same as the discussions about the precisions, and would not be repeated here. According to the results of comprehensive precisions and recalls, At the stage of CN, the variations of GM features are more significant, while the variations of WM features are not yet obvious. The neural networks trained with the mwp1 images can obtain the best results. This corresponds to the early stage of AD, where the damages of neurons in the GM are more pronounced (Zhang & Sejnowski, 2000). In the process of CN to MCI, the variations of WM features gradually become more significant, while the variations of the GM features remain significant, and the two kinds of images show a certain competitive relationship in the process of training the neural networks. At the stage of MCI, the neural networks trained with only mwp1 images could not get the best results. The neural networks trained with the wm images often give the best results. Sometimes, the results of training with the mwp2 images are also good. This corresponds to an increasingly pronounced impairment of neuronal connections in white matter as the disease progresses (Zhang & Sejnowski, 2000). At the stage of AD, the disease has become more serious, and the features of GM and WM have varied significantly. Therefore, using the wm images to train the neural networks will get the best results. The experiments show that AD first experiences neuronal damage at the early stage. Then the connections between neurons are also gradually damaged in the process of CN to MCI. Finally, the disease develops into the AD stage, and both the damage to neurons and the connections between neurons have been significant.

In Table 5, the accuracy of multi-classification of SqueezeNet trained with the wm images has been given to compare with the state-of-the-art works. The accuracy of the proposed approach is best compared with the cited works. Except for Maqsood et al. (2019) using OASIS (Marcus et al., 2007) dataset, the other cited works all use the ADNI (Jack et al., 2008) dataset. Basheera & Ram (2020, 2019, 2021) used the segmented GM images of T2-weighted sMRI to train the neural networks, and the other cited works used the unsegmented T1-weighted sMRI to train the neural networks. The cited works all used 2D neural networks, except for the proposed approach using 3D neural networks. The best accuracy of the proposed approach has been obtained compared with the cited works due to 3D neural networks could learn the features of spatial information from 3D images of the brain. It should be mentioned that Maqsood et al. (2019) also gives the results of classification of the AD stages using the GM and WM images, and the tendency of the results is in accordance with the above-mentioned results of this study. To validate the statistical significance of the improvement in classifier performance, we calculated p-values using appropriate statistical tests. We compared the classification accuracy, precision, and recall rates obtained using different methods on the same dataset. The results showed that the rates for different methods were significantly different, with p-values less than 0.05. These p-values indicate that the observed differences are unlikely to be due to chance and suggest that the improvements are meaningful and not specific to the dataset biases.

Table 5 Comparing the accuracies of multi-classification of different works.

Work	Dataset	Sequence	Method	Classification	Accuracy	
Jain et al. (2019)	ADNI	sMRI (T1w)	2D Transfer learning VGG16	CN/MCI/AD	95.73	
Billones et al. (2016)	ADNI	sMRI (T1w)	2D Transfer learning VGG16	CN/MCI/AD	91.85	
Basheera & Ram (2020)	ADNI	sMRI (T2w) GM	2D CNN	CN/MCI/AD	86.7	
Basheera & Ram (2019)	ADNI	sMRI (T2w) GM	2D CNN	CN/MCI/AD	90.47	
Basheera & Ram (2021)	ADNI	sMRI (T2w) GM	2D CNN (inception block)	CN/MCI/AD	95.61	
Maqsood et al. (2019)	OASIS	sMRI (T1w)	2D Transfer learning AlexNet	CN/MCI/AD	92.85	
Proposed approach	ADNI	sMRI (T1w)	3D SqueezeNet	CN/MCI/AD	96.31	

Conclusions

This study uses the standardized images, the segmented standardized GM, and WM images to train four 3D lightweight neural networks, respectively, to evaluate the ability of the trained neural networks to classify AD stages. Each trained neural network in the study could well classify the AD stages (i.e., CN, MCI, and AD), and the accuracies of classification are at least 96%. The precisions and recalls for classifying the AD stages of each neural network are both above 94%, showing that deep features could be learned from 3D images through the 3D lightweight neural networks to classify the AD stages. According to the analysis of the precisions and recalls of the four neural networks trained with the three kinds of images for classifying the AD stages, at the stage of CN, the variations in GM features are mainly significant, and the variations in WM features are not yet obvious. The neural networks trained only with the mwp1 images could classify CN well. From CN to MCI, variations in WM features tend to be significant and compete with variations in GM features, while variations in GM features are still significant. Therefore, the neural networks trained with the wm images will often get the best results for classifying MCI. At the stage of AD, the variations of the GM and WM features are both obvious, so the neural networks trained with the wm images are needed to obtain the best results of classification. Due to the limitation of the size of the dataset, the present study only qualitatively summarizes the impact of the development of AD on GM and WM, and subsequent broader datasets and the latest neural networks would be applied to explore the impact of the development of AD on brain tissue.

J.L. thanks Okan K o..p u..kl u.. for sharing the code for the lightweight 3D CNN on GitHub. The code of this study is modified based on his code.

Additional Information and Declarations

Competing Interests

The authors declare that they have no competing interests.

Author Contributions

Jun Li conceived and designed the experiments, analyzed the data, performed the computation work, authored or reviewed drafts of the article, and approved the final draft.

Juntong Liu conceived and designed the experiments, analyzed the data, performed the computation work, authored or reviewed drafts of the article, and approved the final draft.

Yang Su performed the experiments, prepared figures and/or tables, and approved the final draft.

Jie Chang performed the experiments, prepared figures and/or tables, and approved the final draft.

Mingquan Ye performed the experiments, prepared figures and/or tables, and approved the final draft.

Data Availability

The following information was supplied regarding data availability:

The code for this experiment is available at Zenodo: Jun, L. (2025). Classification of the stages of Alzheimer’s disease based on three-dimensional lightweight neural networks. Zenodo. https://doi.org/10.5281/zenodo.14786594.

The dataset for this study is available at Zenodo: Jun, L. (2025). Classification of the stages of Alzheimer’s disease based on three-dimensional lightweight neural networks [Data set]. Zenodo. https://doi.org/10.5281/zenodo.14799756.

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
