# Peer review of "Classification of the stages of Alzheimer’s disease based on three-dimensional lightweight neural networks"

_PeerJ Computer Science, doi:10.7717/peerj-cs.2897_

## Round 0.1 · original submission · Major Revisions

Based on the external reviews, a major revision is required. The authors should revise the manuscript accordingly for further consideration.

Reviewer 1 ·

Basic reporting

Reviewer Comments and Suggestions for Improvement
- Replace all abbreviations in the abstract with their full forms (e.g., Alzheimer’s Disease instead of AD) to ensure clarity.
- Clearly state the study's originality and articulate why the chosen methodology is superior to existing methods, emphasizing its specific contributions.
- Highlight the limitations or gaps in existing methods discussed in the literature review and explain how this study addresses these issues.
- Rewrite certain sentences for better flow and readability, ensuring concise and professional phrasing.
- Present the hyperparameters used in the study in a separate table for better accessibility and clarity.
- Emphasize the study's contributions more explicitly, particularly in the abstract, introduction, and conclusion sections.
- Provide more detailed explanations of data preprocessing steps and model parameter configurations.
- Expand the discussion of the study's limitations, particularly regarding the dataset size and the simplicity of the selected models.


By addressing these points, the authors can significantly strengthen the clarity, impact, and comprehensiveness of the manuscript.

Experimental design

no comment

Validity of the findings

no comment

Additional comments

no comment

Reviewer 2 ·

Basic reporting

The manuscript is not professionally written. Although its a good contribution but not presented appropriately. There are numerous terms which are ambiguous. Always remember that when writing a manuscript the language should be simple and self explanatory so that a person from non-technical background can easily understand your contribution.

The Introduction section needs to be re-written. The current contents in the introduction section is relevant to the literature review which is missing in the paper. In introduction there should be contribution mentioned clearly. The objectives and contribution is missing in the introduction section.

Providing results are not enough if that seems different than the explanation in other sections.

Experimental design

The technical terms are not well explained and experimental details are not clearly described. The preprocessing steps are not defined int he paper. Method is not sufficiently defined. Why SqueezeNet is suitable for their work? Why not other networks? What is an architecture of their proposed network? All such important points are not defined clearly in the paper.

The comparison of their work with recently proposed work is not presented in the paper. Paper needs more work to have a mature look overall.

Validity of the findings

Comparison of their work with already established research has not provided. Novelty of their work is not assessed.

Evaluations require another separate section where number of experiment conducted and reported should be clearly mentioned. There is alot of ambiguity in the explanation.

Limitations has not identified int he introduction section, it is not certain what limitation they are intending to address through this paper.

Additional comments

The paper is a good contribution but presentation is very poor. Authors require to rewrite whole paper. Every section needs to be written in simple English with a complete clarity and flow of information through the sections.
Always write paper by keeping in mind those people who are not from the same background as you are. Explain each point, if there is a new term introduce then explain or provide a reference about it. Methodology needs to revise as it is not clearly describe what methodology they used and why they prefer to perform experiments using the method they mentioned.
The obtained results should be compared with the existing work that improves validity of the paper but that is not provided.
This paper has serious presentation concerns that needs to be addressed first.

Annotated reviews are not available for download in order to protect the identity of reviewers who chose to remain anonymous.

Reviewer 3 ·

Basic reporting

the manuscript is clear and easy to read.

Experimental design

The methodology is clear and linear, but there are several missing elements.

1. The manuscript uses lightweight neural networks to avoid overfitting due to the scarcity of data. However, no tests were performed on larger methods. The lack of test and other methods to compare to does not support the authors' claim.

2. In my opinion, a comparison with other approaches should be made. The actual benefit of using GM and WM segmented images cannot be confirmed due to the lack of comparisons.

3. The data collected presents a balancing problem that has not been considered by the authors. This could lead to misclassification and general problems. The authors should comment on this detail.

4. Comparisons with 2D methods should be reported.

Validity of the findings

The lack of comparisons limits the impact of this work.
The lack of comparison with other methodologies makes it difficult to define the novelty of the proposed approach. Also, the benefit of adopting lightweight models is not clear since there is no ablation study on the relationship between model dimensions and performance.

Additional comments

In its current version, the manuscript lacks a strong evaluation to support the claim made by the authors. An in-depth analysis of the proposed method characteristics and a better comparison with the state-of-the-art should be done to improve the paper.

---

## Round 0.2 · Major Revisions

Based on the external review, a major revision is needed.

Reviewer 1 ·

Basic reporting

Reviewer’s Feedback on Revisions

The authors have addressed some of the concerns previously raised; however, several key issues remain unresolved. Please carefully consider the following points to further improve the manuscript:

1. Literature Review and Contextualization:
- The discussion on deep learning applications in medical diagnostics can be strengthened by referencing relevant studies. The following works may be useful:
- Ozdemir, C., & Dogan, Y. (2024). *Advancing brain tumor classification through MTAP model: an innovative approach in medical diagnostics.* Medical & Biological Engineering & Computing, 1–12.
- Ozdemir, C., & Dogan, Y. (2024). *Advancing early diagnosis of Alzheimer’s disease with next-generation deep learning methods.* Biomedical Signal Processing and Control, 96, 106614.
- The latest state-of-the-art techniques should be discussed to highlight recent advancements in the field.

2. Language and Clarity:
- Some grammatical inconsistencies and minor typographical errors persist. A thorough language review would enhance readability and clarity.

3. Novelty and Contribution:
- The introduction provides a clear overview of Alzheimer’s disease (AD) and the importance of early detection. However, the discussion on the novelty of this study compared to prior works should be more explicitly stated.

Experimental design

4. Data Augmentation:
- Consider citing the following study to support the data augmentation methodology:
- Ozdemir, C., Dogan, Y., & Kaya, Y. (2024). *RGB-Angle-Wheel: A new data augmentation method for deep learning models.* Knowledge-Based Systems, 291, 111615.

5. Figures and Tables:
- Some graphs (e.g., Figures 2 and 3) could benefit from clearer labeling and higher resolution for better readability.
- The confusion matrices in Figure 4 should include percentage values in addition to raw counts to improve interpretability.

6. Architectural Design Justification:
- For describing convolution layers, batch normalization, and activation functions, the following references may provide valuable insights:
- Ozdemir, C. (2023). *Classification of Brain Tumors from MR Images Using a New CNN Architecture.* Traitement du Signal, 40(2).
- Ozdemir, C. (2024). *Adapting transfer learning models to dataset through pruning and Avg-TopK pooling.* Neural Computing and Applications, 36(11), 6257–6270.

Validity of the findings

7. Hyperparameter Tuning and Model Training:
- More details on hyperparameter tuning, data augmentation techniques (if any), and dropout rates used in training should be provided. Specifically, a detailed table summarizing the hyperparameters used should be included.

8. Preprocessing and Segmentation Verification:
- The image preprocessing pipeline using CAT12 is explained well; however, additional details on how segmentation accuracy was verified are necessary. Was manual inspection or an automated validation approach used?
- The justification for choosing certain preprocessing techniques over others should be elaborated (e.g., why CAT12 instead of alternative segmentation tools?).

9. Statistical Significance and Validation:
- Confidence intervals or statistical significance tests (e.g., McNemar’s test or p-value) are missing. Including these would strengthen the claim that the improvements are meaningful rather than dataset-specific biases.

10. Justification of Methodology:
- The choice of preprocessing techniques, network architectures, and training parameters should be justified more comprehensively.

11. Model Interpretability and Biomarkers:
- The study lacks a discussion on model interpretability. How do the learned features correlate with known biomarkers of AD?
- Techniques such as Grad-CAM or SHAP should be used to visualize important brain regions and provide insights into model decision-making.


Conclusion:
While the manuscript presents promising findings, the aforementioned points need to be addressed to improve the scientific rigor and clarity of the study. Strengthening the statistical validation, justifying methodological choices, and enhancing interpretability would significantly improve the manuscript’s impact and contribution to the field.

Annotated reviews are not available for download in order to protect the identity of reviewers who chose to remain anonymous.

---

## Round 0.3 · accepted · Accept

Based on the reviewer's feedback and authors' response, the manuscript can be accepted.

Reviewer 1 ·

Basic reporting

Accept

Experimental design

Accept

Validity of the findings

Accept

Additional comments

Accept